# Multilingual Generation and Answering of Questions from Texts and Knowledge Graphs

**Kelvin Han** and **Claire Gardent**
CNRS/LORIA and Université de Lorraine
{huiyuan.han,claire.gardent}@loria.fr

## Abstract

The ability to bridge Question Generation (QG) and Question Answering (QA) across structured and unstructured modalities has the potential for aiding different NLP applications. One key application is in QA-based methods that have recently been shown to be useful for automatically evaluating Natural Language (NL) texts generated from Knowledge Graphs (KG). While methods have been proposed for QG-QA across these modalities, these efforts have been in English only; in this work, we bring multilinguality (Brazilian Portuguese and Russian) to multimodal (KG and NL) QG-QA. Using synthetic data generation and machine translation to produce QG-QA data that is aligned between graph and text, we are able to train multimodal, multi-task models that can perform multimodal QG and QA in Portuguese and Russian. We show that our approach outperforms a baseline which is derived from previous work on English and adapted to handle these two languages. Our code, data and models are available at https://gitlab.inria.fr/hankelvin/multlingual_kg-text_qgqa.

## 1 Introduction

The ability to generate and answer questions from both Knowledge Graphs (KG) and Natural Language (NL) text is useful in a number of ways. It permits easing users' access to the knowledge contained in KGs without having to master complex query languages, since an NL query from a user may be directly applied to KG information to derive the answer. It also allows for questions to be generated and answered from both the open domain information contained in NL text and the factual knowledge contained in KGs/knowledge bases, thereby widening the pool of information that is interrogable. And it is useful also for verifying the consistency of information between modalities. In particular, (Rebuffel et al., 2021) recently showed that multimodal KG/NL Question Generation (QG)

and Question Answering (QA) can be used for assessing the semantic consistency between an input KG graph and a generated English text, thereby providing a reference-less metric to measure the quality of KG verbalisers (RDF-to-Text models). The intuition is that, for a match between an input and its output, the answers that are extracted for a given set of questions from the input should be consistent with the answers that are extracted from the output (for the same set of questions).

One key challenge to building multimodal QG-QA however, is the lack of annotated QG-QA data that is aligned between modalities that is available for training and evaluation of such models. Furthermore, to be useful for RDF-to-Text output evaluation, a multimodal QG-QA approach has to be robust to surface variations since the answer to a question likely has different surface forms in the input graph and the output text. Its QG and QA also have to be cross-lingual so that questions from multilingual texts can be generated and answered from English-centric KGs. Additionally, the set of questions generated for a given (graph, text) pair should be sufficiently large and the model(s) should also perform QA with accuracy and consistency, so as to be able to provide a fair assessment of the semantic consistency between graph and text.

In this work, we investigate the generation and answering of questions from graph and from text for multiple languages besides English. We present models enabling this for Russian and in Brazilian Portuguese. In addition, we apply our approach to WebNLG, a dataset of (graph, text) pairs that is used to train RDF-to-Text models, and we show that our approach extends to (Rebuffel et al., 2021)'s reference-less, semantic adequacy metric for RDF-to-Text models for these two languages.

We make the following contributions. First, we create training data where questions are aligned with both their graph and their text answers, allowing the training of models that show cross-modal

consistency: i.e. for a given question, the answers obtained from a graph and from its semantically equivalent text are consistent.

Second, we derive silver training data from English using Machine Translation (MT) and we demonstrate that the resulting multimodal, Portuguese and Russian QG-QA models trained on them have high internal consistency: in most cases, applying a Russian or Portuguese question generated from a graph to the same graph returns an answer that matches the graph entity that the question was originally conditionally-generated from.

Third, we show that our approach has much larger question coverage than a baseline adapting (Rebuffel et al., 2021)'s approach to our target languages. Finally, to overcome the lack of any gold QA/QG data that is aligned between text and graph in these two languages, we designed our evaluation suite to include multiple internal, cross-modal, cross-approach and external checks. These checks also demonstrate that our approach significantly improves on the baseline.

## 1.1 Terminology and notations

| | |
|---|---|
| *En* | English |
| *PtBr* | Brazilian Portuguese |
| *Ru* | Russian |
| $g$ | (or *KG graphs/graphs*). A subgraph of the Wikidata KG (Vrandečić and Krötzsch, 2014); comprised of a set of triples (also called facts) of the form ⟨subject, predicate, object⟩ in English. |
| $t$ | (or *NL texts/texts*). A text in English/Portuguese/Russian |
| $X$ | The context of a question in one modality (text or graph) |
| $X'$ | Semantically equivalent to $X$ in the other modality |
| $q$ | A question |
| $\vec{q}$ | A collection of questions |
| $a_X$ | An answer in $X$ (a graph answer ($a_g$) is either a subject or an object entity in $g$; a text answer ($a_t$) is a span in $t$) |
| $g'$ | A subgraph of $g$ corresponding to a $q$ and its answer |
| $nf$ | The number of facts related to a given $q$ (i.e. the size of its corresponding subgraph $|g'|$ ) |
| $t^{PtBr}$ | A text in Portuguese (the superscript distinguishes languages) |

## 2 Approach

Since there is no cross-modal QG/QA data for Brazilian Portuguese and Russian, we approach the task by first creating the data necessary for training multimodal QG-QA models for English. We then machine translate this data to create training data for Brazilian Portuguese and Russian. Finally, we use this automatically translated data to train multimodal (KG/NL), multi-task (QG-QA) models for these two languages. The following outlines our approach and details are provided in the subsequent sections.

- **Creating Question/Answer Data for pairs of English Texts and Wikidata Knowledge Graphs.** We first create a large synthetic dataset (Section 3) of QA pairs for graphs and texts in English as leveraging existing resources (datasets and models) already developed for that language allows us to obtain QA pairs at scale. We do this by applying off-the shelf QG and QA models to texts from KELM (Agarwal et al., 2021), a large dataset of (Wikidata graph, English text) pairs. We call this data Q-KELM$^{En}$.

- **Learning Controllable QG Models for English Texts and Wikidata Knowledge Graphs.** Using the Q-KELM$^{En}$ dataset for training, we learn two controllable graph- and text-based QG models which allows for multiple, varied questions, of different graph sizes and question types, to be generated from the same (graph, text) pairs. This is essential as it is through controllable generation with these models that we can significantly increase the QA/QG training data coverage (Section 7.2).

- **Creating Question & Answer Data for the WebNLG Dataset (from both English Texts and Graphs).** By applying our controllable QG models to the (graph, text) pairs of the WEBNLG (Gardent et al., 2017) dataset, we create a large dataset of (question, answer) pairs from texts and graphs that can be used to train models used in verifying the semantic match between WEBNLG graphs and texts generated from these graphs.

- **Learning Multimodal, Multi-task QG-QA Models for Brazilian Portuguese and for Russian.** By further using MT, heuristics and quality filters (Section 5), we obtain silver-aligned, in-domain QA pairs that enable us to train QG-QA models for WEBNLG graphs and their texts in Portuguese or Russian.

- **Testing on WEBNLG and with an independent QA model.** We apply our Portuguese and Russian, multimodal QG-QA models to WEBNLG Portuguese and Russian evaluation data and compute correlation (Section 7.4) with human judge-

ment of semantic adequacy (i.e. Does a generated text match the semantic content of its input graph?). We also apply the generated questions to a retrieval-based multilingual QA model (Section 7.5) to further verify the quality of the QG (i.e. How well can the generated questions be answered by an external model?).

We show that our approach brings substantial improvements over the baseline, to question coverage, QA consistency, correlations with human judgments of semantic adequacy as well as answerability by a retrieval-based multilingual QA model.

## 3 Data

- WEBNLG has 38,872 (graph, English texts) pairs where each graph is a Wikidata graph and each English text was crowdsourced to match the input graph.[1]

- WEBNLG$^{Ru}$: (Shimorina et al., 2019) created a Russian version of WEBNLG by machine translating 16,522 English texts from the original WEBNLG dataset, followed by crowdsourced post-editing. This dataset was used in the 2020 WebNLG Challenge (Castro Ferreira et al., 2020).

- WEBNLG$^{PtBr}$: Similarly, (Almeida Costa et al., 2020), created a Brazilian Portuguese version of the WEBNLG test set by using MT and a pool of native Portuguese speakers for post-editing.

- KELM consists of 15M (Wikidata, English Texts) pairs. The English texts were synthetically generated from Wikidata graphs using a T5 pre-trained model that was itself fine-tuned on TekGen, a large dataset of (graph, text) pairs that was created using distant supervision.

We reserve the test sets of WEBNLG$^{PtBr}$ and WEBNLG$^{Ru}$ — 1,606 $(g, t^{PtBr})$ and 1,102 $(g, t^{Ru})$ pairs respectively — for evaluation as the parallel $(g, t)$ in these test sets means that if a question can be answered by a text (or graph), it can also be answered by the graph (or text) that it is paired with — this allows us to test cross-modal QG and QA for consistency (Section 7.3). We use the training portion of WEBNLG to produce our QA/QG datasets, which is done in two phases and which we describe in the following sections and illustrate in Figure 1.

---

[1]In WEBNLG, the graphs are from the DBpedia KG. Here we use a version where some of the DBPedia graphs have been mapped to Wikidata (Han et al., 2022), or else removed of underscores and camelcase to align with the Wikidata format.

## 4 Creating QA/QG Data for English

Our first phase generates synthetic (question, answer) pairs for WEBNLG graphs and English texts in three main steps as follows.

**Step 1: Synthetic QA data** In the first step, we derive $(t, a_t, q)$ triplets from KELM texts[2] using off-the-shelf models for textual QG and QA.[3] We first use the QG model to generate questions from texts. Next we use the QA model to generate answers from the resulting (text, question) pairs. Using a set of heuristics, we then align (i) each text answer $a_t$ to an entity in $g$ giving us the graph answer $a_g$ and (ii) each question to the subgraph $g'$ of $g$ matching its content, which in turn allows us to obtain the question size, $nf$. The set of generated questions is then filtered to ensure the quality of the silver data; this includes posing each question to a second deepset textual QA model for English and only accepting questions which both QA models found answerable and whose answers are the same or overlap. Questions whose text answer could not be matched to a graph answer were also removed. This gives us an initial synthetic QA/QG dataset of $(g, a_g, t, a_t, nf, q, g')$ tuples, which we call Q-KELM$^{En}$.

**Step 2: English Text/KG QG models** Using Q-KELM$^{En}$, we train two QG models that can controllably generate simple (one KG graph fact) and complex (>1 KG graph fact) questions from either text or graph. The controls we use are: the size of the question in terms of KG triples ($nf$), and plausible question types (e.g., who, what, when, where, which, how (to)) in the case of simple questions generation. These controllable QG models enable our generation of wide-coverage in-domain QG-QA data that also allows us to generate multiple questions per input (see next point and Section 6).

---

[2]We used a version of KELM filtered for $(g, t)$ pairs where $g$ has between 2 and 5 triples (as larger sizes lead to unnatural questions). We also filtered out the KELM $(g, t)$ pairs (i) whose properties in $g$ were not found in the Wikidata SPARQL endpoint or have a functional nature, i.e. containing terms such as 'image of', 'instance of human', 'list of', 'disambiguation' which gives rise to superfluous $t$ in KELM; and (ii) $t$ that do not have a high fidelity with their $g$, using a similarity measure (Scao and Gardent, 2023) trained via contrastive loss on RDF graph-text pairs.

[3]https://huggingface.co/valhalla/t5-base-e2e-qg, a T5-base QG model fine-tuned on SQuAD 1.0 data, as well as https://huggingface.co/deepset/roberta-base-squad2 and https://huggingface.co/deepset/deberta-v3-base-squad2 which are RoBERTa/DeBERTa-base QA models that are both fine-tuned on SQuAD 2.0.

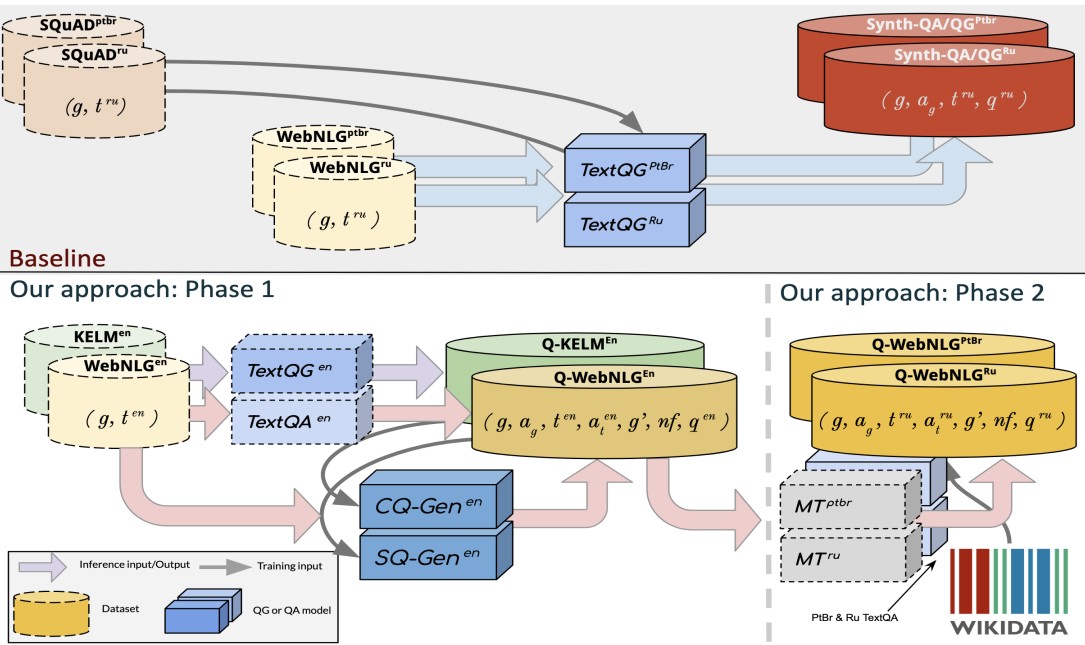

Figure 1: **Comparing our approach against the baseline.** Using KELM and controllable QG increases coverage. Aligning questions with contexts and answers across modalities improves consistency. Machine translation provides multilingual training data.

**Step 3: In-domain training data** Finally, these two models are applied to the training portion of WEBNLG. By cycling through the set of possible graph and text answers, and leveraging the controls we introduced in Step 2 (question sizes and question types), we ensure wide coverage of the resulting set of questions.

**Answerability, consistency and alignment** We add an answerability+consistency filter on the questions generated from either graph or text of WEBNLG, by posing them to the two deepset textual QA models — we then keep only questions where both QA models return an answer which (i) has a confidence score $\geqslant 0.7$, (ii) shares at least a token overlap with the other model's answer, and (iii) has a token overlap with the answer used to condition QG (i.e. a text answer for questions from text; a graph answer for questions from graph). In this way, for all questions that are generated from a graph, we ensure that they have a matching graph and text answer. We call the resulting dataset — of graph- and text-based (question, answer) pairs, Q-WEBNLG$^{En}$.

## 5 Creating QA/QG Data for Russian and Portuguese

In the second phase, we leverage MT and multilingual KG entity labels from Wikidata to trans-

form Q-WEBNLG$^{En}$ into Portuguese and Russian versions.

We describe the process for Portuguese first. We translate both the English texts of WEBNLG training data and the set of questions from Q-WEBNLG$^{En}$ using a T5 English-Portuguese translation model[4]. We filter these translations using checks (details in Appendix A.2) that include back-translation and keeping only those whose automatic scores are above a cut-off. Since we machine translate the questions from English, the semantics of the questions might be affected; as such for each translated question $q$, we then use a textual QA model trained on Portuguese SQuAD data to obtain the answer to that question from the (automatically translated) Portuguese text ($a_t^{PtBr}$) and we use heuristics (Appendix A.2.1) to align this text answer to an entity $a_g$ in the matching graph[5]. We discard any QA pair for which $a_g^{PtBr}$ is different from the graph answer $a_g^{en}$ associated with the original English question (the question $q$ is a translation of).

We do the same for Russian, except that we do not translate WEBNLG training data to Russian as it is already available (Shimorina et al., 2019). To translate the Q-WEBNLG$^{En}$ questions into Rus-

---

[4] https://huggingface.co/unicamp-dl/translation-en-pt-t5

[5] Recall that in WEBNLG, each text is paired with a matching graph with semantically equivalent content.

| nf | Q-K$^{En}$ | Q-W$^{En}$ | Q-W$^{PtBr}$ | Q-W$^{Ru}$ |
|---|---|---|---|---|
| 1 | 44,464 | 19,467 | 7,557 | 3,250 |
| 2 | 341,082 | 61,346 | 15,463 | 8,736 |
| 3 | 234,170 | 25,170 | 5,446 | 3,612 |
| 4 | 22,607 | 13,918 | 2,411 | 2,004 |
| 5 | 7,031 | - | - | - |
| TOTAL | 1,149,354 | 119,901 | 30,877 | 17,602 |

Table 1: **Data Statistics.** Number of questions in the QA datasets; $nf$: the size of the question (no. of facts).

sian, we use NLLB-200-1.3B[6] (NLLB Team et al., 2022), an MT model achieving state-of-the-art for 200 languages.

We call the resulting QA datasets, Q-WEBNLG$^{PtBr}$ and Q-WEBNLG$^{Ru}$. Table 1 summarises their statistics and example instances of the data can be seen in Table 7 in Appendix A.

## 6 Multimodal multi-task QG-QA model

We train monolingual models (i.e. one each for Portuguese and Russian) to limit the effect of cross-lingual transfer during training (i.e. only English-Portuguese or English-Russian when generating and answering with graph). Each language-specific model — which is fine-tuned from the public check-point of the 300M-parameter mT5-small (Xue et al., 2021) for 10 epochs (to be comparable with the baseline) — is trained in a multimodal (graph and text), multi-task (QG and QA) manner, where each training batch contains a mix of all the tasks, i.e. Text QG, Text QA, KG QG and KG QA. This gives us a single unified model that can generate and answer questions from text and from graph.

It takes approximately 20 hours to fine-tune one of our language-specific multimodal multi-task models using a single Nvidia A40 GPU.

**Maximising Coverage** To maximise coverage, we train the QG model to generate multiple questions from the same input and we extend the set of possible sources for a (question, answer) pair thereby facilitating question answering.

For QG, we gather into a set the questions generated from each $X$ ($t$ or $g$) in Q-WEBNLG$^{PtBr}$/Q-WEBNLG$^{Ru}$, and add to it questions that were generated from other contexts whose semantic content is contained in $X$. QG coverage is then maximised by gathering all $nf$-sized questions that shared the same answer in this set — by doing so, each of

our QG training instance can then contain multiple questions, enabling the generation of multiple questions from a given $(X, a_X, nf)$ input.

For QA, we associate each $(q, a_X)$ pair from a given $X$ to other contexts in the data that encompass $X$ to give $(X, a_X, \overrightarrow{q})$; every $(X, a_X, q)$ triplet in this set is then created as a QA training instance, thereby allowing QA coverage to be increased.

**Handling Unanswerable Questions.** To allow our model to abstain from an answer if the question cannot be answered from the context, we use two strategies (details in Appendix B.1) to obtain negative unanswerable $(q, \neg X)$ pairs.

## 7 Evaluation and results

### 7.1 Baseline: single-task models

| Dataset | Portuguese | Russian |
|---|---|---|
| SQuAD | 118,678 | 50,364 |
| WebNLG | 125,957 | 62,007 |
| TOTAL | 244,635 | 112,371 |

Table 2: **Baseline** training data. Number of $(t, a_t, q)$ triplets in the obtained datasets for each language.

The baseline we compare against comprises four different models for each language and is similar to the approach described for the Data-QuestEval metric (Rebuffel et al., 2021) for English, which they have shown to be useful for reference-less evaluation of English RDF-to-Text models. First, textual QG and QA models are trained on SQuAD data for Portuguese and Russian.[7] The textual QG models are then applied to the Portuguese and Russian texts of the WEBNLG training data; whereby the entities from the graph that is paired with each text are used to condition QG. This provides $(g, a_g, q)$ triplets that can be used for training the KG QG and KG QA models. Table 2 contains statistics about the training data for the baseline. To remain in the same model paradigm as (Rebuffel et al., 2021), every model here is also fine-tuned from mT5-small for 10 epochs.

[6]https://huggingface.co/facebook/nllb-200-distilled-1.3B

[7]For Portuguese we use a version of SQuAD v1.0 (Rajpurkar et al., 2016) we understand is produced with MT and post-edited by native speakers, available at https://huggingface.co/datasets/ArthurBaia/squad_v1_pt_br; for Russian we use the SberQuad (Efimov et al., 2020) dataset.

## 7.2 Evaluating Question Coverage

We use two measures to assess question coverage. We first compare the number of unique questions generated by each approach on the WEBNLG$^{PtBr}$/WEBNLG$^{Ru}$ test sets. We further use BERTScore (BSc) (Zhang et al., 2020) to assess their semantic overlap by taking one approach's question for a given $X$ as prediction and the other's generated questions for the same $X$ as multi-references. The intuition is that if approach A scores higher with approach B's questions as references than vice-versa, A's questions are "contained" in B's and conversely, B has wider semantic coverage.

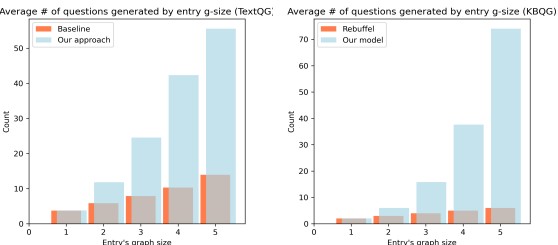

Figure 2: **Portuguese: Comparative QG coverage — Baseline vs. Our Approach.** Number of generated questions is much higher for our approach across both modality and input size.

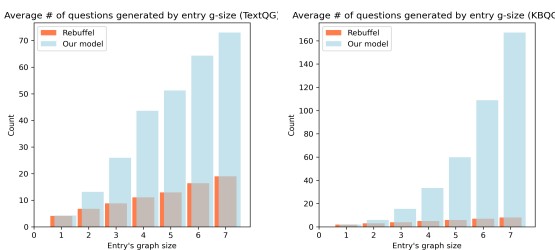

Figure 3: **Russian: Comparative QG coverage — Baseline vs. Our Approach.** Number of generated questions is also much higher for our approach across both modality and input size.

■ ***Wider $q$ coverage*** *Our approach has a substantially higher question coverage (up to 4x more for Portuguese and nearly 7x more for Russian) than the baseline (Figure 2-3). In terms of semantic coverage (Table 3), we outperform the baseline in all modalities for Portuguese; we also outperform in graph for Russian, and are on par in the text modality there.*

## 7.3 Evaluating QA Consistency

In what follows, the answer $a_X$ that is used to condition the generation of $q_X$ is referred to as the

| Modality | As reference | Portuguese | Russian |
|---|---|---|---|
| Text | Baseline | 83.1 | **81.3** |
| | Ours | **85.0** | **81.3** |
| Graph | Baseline | 80.7 | 75.9 |
| | Ours | **84.6** | **80.1** |

Table 3: Avg BSc, where questions generated by System A for a given $g$ or $t$ are scored against the set of generated questions by System B for the same $g$ or $t$.

ground truth (GT). $\hat{a_X}$ denotes a generated answer from context X. We use superscripts distinguish outputs from different models — for e.g., $\hat{a}_X^A$ is the answer derived from input context $X$ by model $A$.

We evaluate the multimodal QG-QA models by computing three consistency metrics that consider the various answers that a question $q$ can be associated with: $a_X$, the ground truth; $\hat{a}_X$, the answer generated from source $X$ (e.g., a text); and $\hat{a}_{X'}$, the answer generated from the other modality $X'$ (e.g., a graph). *Internal Same-mod (GT)* compares $\hat{a}_X$ with the ground truth answer $a_X$, indicating the approach's self-consistency. *Internal X-mod (GT)* compares $a_X$ with $\hat{a}_{X'}$ the answer derived from the other modality. *Internal X-mod (Gen Ans)* compares $\hat{a}_{X'}$ and $\hat{a}_X$, the answers from each modality.

By also posing the questions generated by one approach to the other approach's QA in a cross approach manner, we gain an external indication of their QG-QA capabilities. For this, we examine the two answers that approach B can generate ($\hat{a}_X^B$ and $\hat{a}_{X'}^B$) when given $q_X^A$, a question generated by A. We do this on three levels: (i) *X-Appr Same-mod (GT)* compares $\hat{a}_X^B$ against $a_X^A$ on the same modality that $q_X^A$ came from; (ii) *X-Appr X-mod (GT)* compares B's generated answer with the GT answer across modalities (i.e. $\hat{a}_{X'}^B$ vs $a_X^A$); and (iii) *X-Appr X-mod (Gen Ans)* where $\hat{a}_{X'}^B$ is compared against $\hat{a}_X^A$. A graphical overview of these comparisons is in Figure 4.

Both approaches usually generate a different number of questions for a given $X$; therefore, to ensure a fair evaluation, when an Approach A generates more questions for $X$, we randomly sample from its set as many questions that Approach B generates for $X$. We also added a self-consistency filter removing those questions that are unanswerable from their own context. Settings for these are in Appendix B.3.

As the token F1 metric commonly used in extractive textual QA cannot account for lexical variation

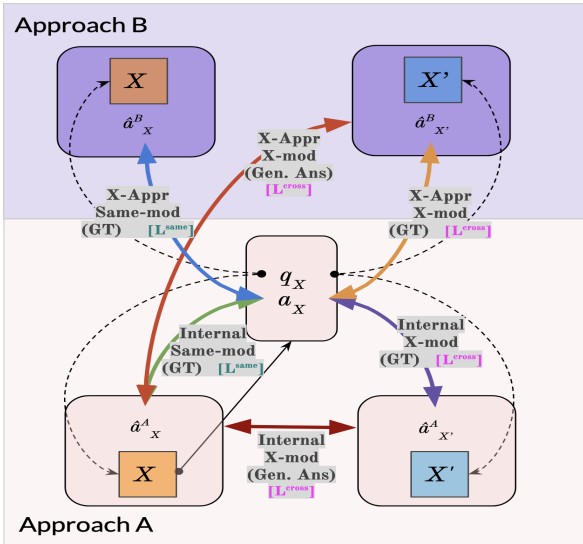

Figure 4: **QA Accuracy**. Bold lines denote QA comparisons within/between modalities and/or approaches. Dotted arrows indicate the context $X$ or $X'$ that the question ($q_X$) is posed against to obtain the answers. $L^{cross}$: cross-lingual answer comparison (e.g. PtBr/Ru against En); $L^{same}$ denote comparison in the same language.

present in cross-modal QA, we follow (Rebuffel et al., 2021)'s use of BERTScore (BSc) for evaluation; however, we also computed token F1 and exact match scores for verification, and these are provided in Tables 8-9 in Appendix C.1.

Table 4 show the results of the QA consistency tests for Portuguese and Russian. We observe similar trends for both languages, though the BScs for Russian are noticeably lower under the cross-modal (and cross-lingual too, since $a_g$ is in English) settings. This is likely due to (i) the smaller size of the training data for Russian (Table 1), (ii) more frequent transliteration of foreign names into Cyrillic, and (iii) potentially, the freer word order in Russian which relies on case instead of S-V-O order to mark the subject and object (as it is for English and Portuguese).

■ *More self-consistent QA* *Our approach consistently leads to higher scores compared to the baseline in the* Internal *comparisons for all languages, showing that multimodal, multi-task training using silver aligned data (even at much smaller scale — nearly 4 times less for Portuguese and more than 3 times less for Russian; cf. Table 1 and 2), improves QA self-consistency.*

■ *More consistent cross-modal QA* *We also improve by between 8.9 (29.7 over 20.8, Russian) to 22.9 (65.6 over 43.7, Portuguese) BSc*

**Portuguese**

| QG
QA | Internal
Baseline
Baseline | Ours
Ours | X-Appr
Baseline
Ours | Ours
Baseline |
|---|---|---|---|---|
| *Self (GT)* | | | | |
| T → T | $86.3_{(\pm 0.01)}$ | $\mathbf{94.7}_{(\pm 0.14)}$ | $73.6^{(-12.7)}_{(\pm 0.10)}$ | $60.1^{(-34.6)}_{(\pm 0.25)}$ |
| G → G | $85.4_{(\pm 0.03)}$ | $\mathbf{96.8}_{(\pm 0.05)}$ | $58.2^{(-27.2)}_{(\pm 0.12)}$ | $61.0^{(-35.8)}_{(\pm 0.53)}$ |
| *X-mod (GT)* | | | | |
| G → T | $43.7_{(\pm 0.15)}$ | $\mathbf{65.6}_{(\pm 0.37)}$ | $47.0^{(+3.3)}_{(\pm 0.11)}$ | $47.8^{(-17.8)}_{(\pm 0.30)}$ |
| T → G | $40.0_{(\pm 0.08)}$ | $\mathbf{62.9}_{(\pm 0.46)}$ | $45.7^{(+5.7)}_{(\pm 0.12)}$ | $37.8^{(-25.1)}_{(\pm 0.35)}$ |
| *X-mod (Gen Ans)* | | | | |
| G → T | $39.9_{(\pm 0.14)}$ | $\mathbf{67.1}_{(\pm 0.34)}$ | $42.5^{(+2.6)}_{(\pm 0.09)}$ | $47.7^{(-19.4)}_{(\pm 0.30)}$ |
| T → G | $40.0_{(\pm 0.08)}$ | $\mathbf{65.0}_{(\pm 0.37)}$ | $44.3^{(+4.3)}_{(\pm 0.11)}$ | $39.0^{(-26.0)}_{(\pm 0.34)}$ |

**Russian**

| QG
QA | Internal
Baseline
Baseline | Ours
Ours | X-Appr
Baseline
Ours | Ours
Baseline |
|---|---|---|---|---|
| *Self (GT)* | | | | |
| T → T | $84.6_{(\pm 0.02)}$ | $\mathbf{89.8}_{(\pm 0.11)}$ | $57.8^{(-26.8)}_{(\pm 0.10)}$ | $57.2^{(-32.6)}_{(\pm 0.18)}$ |
| G → G | $79.4_{(\pm 0.03)}$ | $\mathbf{96.2}_{(\pm 0.13)}$ | $48.9^{(-30.5)}_{(\pm 0.10)}$ | $57.1^{(-39.1)}_{(\pm 0.53)}$ |
| *X-mod (GT)* | | | | |
| G → T | $20.8_{(\pm 0.04)}$ | $\mathbf{29.7}_{(\pm 0.26)}$ | $24.5^{(+3.7)}_{(\pm 0.06)}$ | $20.6^{(-9.1)}_{(\pm 0.21)}$ |
| T → G | $20.2_{(\pm 0.06)}$ | $\mathbf{30.0}_{(\pm 0.08)}$ | $17.3^{(-2.9)}_{(\pm 0.03)}$ | $21.5^{(-8.5)}_{(\pm 0.06)}$ |
| *X-mod (Gen Ans)* | | | | |
| G → T | $18.6_{(\pm 0.04)}$ | $\mathbf{31.1}_{(\pm 0.27)}$ | $22.8^{(+4.2)}_{(\pm 0.06)}$ | $20.3^{(-10.8)}_{(\pm 0.23)}$ |
| T → G | $18.4_{(\pm 0.07)}$ | $\mathbf{30.8}_{(\pm 0.10)}$ | $15.0^{(-3.4)}_{(\pm 0.03)}$ | $22.2^{(-8.6)}_{(\pm 0.06)}$ |

Table 4: **Consistency Results**. Average of BScs between answers. In subscripts are std. dev. across 5 random runs; in superscripts are the difference between X-Appr and Internal, the differences provides a meaningful comparison between the baseline and our approach since each of their QA performances are different. Whenever our QA is used, the drop in performance is reduced.

*over the baseline in* Internal X-mod (GT). *This is particularly pertinent for the Data-QuestEval metric, as the metric is computed by comparing the generated answer $a_{X'}$ against the GT answer $a_X$. Our approach also always leads to gains in* Internal X-mod (Gen Ans) *over* Internal X-mod (GT), *which does not happen for the baseline. This means that our QG-QA generates answers that are more consistent between both modalities. This improvement could have come at the expense of the QG-QA's performance vis-a-vis the GT answer (i.e.* Internal X-mod (GT)), *but this is not the case for our approach, which further validates our multimodal multi-task training to give*

*more consistent cross-modal QG-QA.*

■ *QG-QA externally validated  Whenever the baseline's QA is used to answer questions generated by our approach, QA accuracy drops significantly less (in some cases, there is a gain) than vice-versa (Table 4). This could be predominantly the result of either (i) better QG by our approach; or (ii) better QA by the baseline — however, our approach's stronger performance in all the Internal, X-mod and X-Appr comparisons suggests that the former is the likely factor. This gives an indication of the quality of the questions generated by our approach over the baseline.*

■ *QA consistent over $q$ size  Finer-grained analysis (Table 10 and 11 in Appendix C.2) shows that our approach's QA performance is consistent across $nf$, i.e. it is also generating and answering questions of consistent quality across questions of different "sizes" (i.e. simple and complex).*

## 7.4  Data-QuestEval metric

The QA consistency tests above reflect a "gold" setting where both modalities (texts and graphs) are semantically aligned. To evaluate whether our approach brings improvements to settings where this does not hold (i.e. Can it generate and answer questions across modalities where one modality has missing, or additional, information vis-a-vis the other?), we compared it against the baseline when used to compute the Data-QuestEval metric.

We do this by comparing the metric's resulting correlation with human judgments of semantic quality when using each approach. For this, we used the sampled set of system submissions to the Russian RDF-to-Text task in the 2020 WebNLG Challenge, together with their human ratings[8], comprising 660 generated texts from six submissions.[9] Since the texts here were machine-generated given a WEBNLG graph input, these texts may contain errors or are ill-formed, therefore the approach giving a higher correlation with the human ratings indicates that it is better able to answer questions that cannot be answered by the other modality (i.e. detect a difference in information content).

■ *Better correlations with human judgments  Using our approach to compute the Data-QuestEval metric leads to a gain of more than 12 points in its correlation with human judgments (Table 5). Together with the more consistent internal and cross-modal QA above, this shows that our approach leads to more robust QG-QA. It also indicates that QA-based reference-free evaluation methods like Data-QuestEval can be improved upon with wider QG coverage.*

| Baseline | Our Approach |
|---|---|
| 16.4 (4.78E-06) | **28.7** (4.21E-16) |

Table 5: **Correlations (Pearson's $r$) with human judgments.** The baseline vs our approach used to compute the Data-QuestEval metric. All ($p$-values) ≪ 0.001.

## 7.5  Multilingual Retrieval-based QA

For a further verification of the QG abilities of the baseline and our approach, we also posed the questions generated by each (the same ones as in Table 4) to mGEN (Asai et al., 2021). This is a multilingual retrieval-based QA model that was fine-tuned from a mT5-base (Xue et al., 2021) checkpoint to generate the answer to a question given a collection of retrieved input contexts.[10]

| **Portuguese** | | |
|---|---|---|
| | **External Retrieval-QA** | |
| QG | Baseline | Ours |
| QA | mGEN | mGEN |
| | *Self (GT)* | |
| T → T | 71.4 ($\pm 0.07$) | **78.0** ($\pm 0.58$) |
| G → G | 57.7 ($\pm 0.13$) | **72.8** ($\pm 0.48$) |

| **Russian** | | |
|---|---|---|
| | **External Retrieval-QA** | |
| QG | Baseline | Ours |
| QA | mGEN | mGEN |
| | *Self (GT)* | |
| T → T | 65.9 ($\pm 0.13$) | **68.5** ($\pm 0.18$) |
| G → G | 66.9 ($\pm 0.15$) | **76.0** ($\pm 0.77$) |

Table 6: **QA consistency (BSc)** comparing the QG of Baseline and Ours, using mGEN for QA.

■ *Better answerable questions  Compared to the baseline, our approach's questions were answered*

---

[8] https://github.com/WebNLG/challenge-2020
[9] We use the ratings of Data Coverage, Relevance and Correctness (summing their normalised scores). Since the outputs for the challenge's human evaluations were sampled in a random stratified manner, we computed correlation using Pearson's $r$.

[10] We use the version of the code and weights that was released as part of the MIA-2022 Shared Task (Asai et al., 2022). See https://github.com/mia-workshop/MIA-Shared-Task-2022/tree/main.

*better by mGEN — we see (Table 6) an increase of between 2.6 BSc for TextQG (Russian) and 15.1 BSc (Portuguese) in answer accuracy, providing a further independent verification of our approach's QG capabilities.*

## 8 Related work

**Joint QG & QA and data**    Existing work on joint QG & QA (Wang et al., 2017; Duan et al., 2017; Lyu et al., 2021; Luo et al., 2022) have mostly focused on the text modality and in English only. Where more than one modality or multiple languages are considered, these have usually been for QA alone. This is partly because available annotated multilingual QA/QG datasets (Lewis et al., 2020; Artetxe et al., 2020) are limited in size and primarily reserved for testing, as well as built for one modality only. While there exists moderate-sized KG QA datasets, including the long-running QALD (Usbeck et al., 2023) and LC-QuAD (Dubey et al., 2019) KG QA challenges, these are still of a limited size (and coverage). Furthermore, none of these QA/QG datasets have alignment of QA pairs across modalities, as well as across languages. Notably, since open KGs such as Wikidata (though large and with wide and growing fact coverage) remain English-centric, the setting we examine has to involve cross-linguality when carrying out QG and QA in the graph modality.

**Bridging modalities**    Early efforts to bridge QA between modalities focused on supplementing limited KG coverage by extracting relational information from more abundant texts. For instance, (Fader et al., 2014; Das et al., 2017) leveraged both structured (KB, tables, lists etc) and unstructured (text) information and used information extraction methods such as OpenIE (Banko et al., 2007) and UniversalSchema (Yao et al., 2012) so as to employ semantic parsing- or rules-based KG QA methods. More recent work instead casts structured information as text to access their knowledge through textual QA methods. (Agarwal et al., 2021) constructed KELM as a verbalisation of a large KG (Wikidata) to add to a retrieval LM corpus, obtaining performance improvements on benchmark QA datasets. (Oguz et al., 2022) obtain improvements by adding Wikipedia tables and lists to the data mix. In a similar vein as (Agarwal et al., 2021), (Zhang et al., 2023) verbalise a multilingual KG QA dataset to utilise textual QA methods.

**Synthetic data**    Other work has investigated the use of synthetically-generated data with round trip filtering techniques and shown improved textual QA performance (Alberti et al., 2019; Puri et al., 2020; Kwiatkowski et al., 2019). (Riabi et al., 2021; Agrawal et al., 2023) have also examined multilingual synthetic QA/QG data generation. Similarly, we used data augmentation and round trip filtering to improve generalisation; however, these other work are aimed at improving textual QA only; unlike ours which aims to improve multilingual QG and QA jointly, which is also cross-modal for text and graph while also ensuring wide QG coverage.

**QA-based evaluation of texts**    Our work is most similar to (Rebuffel et al., 2021) who use an approach like our baseline to show that QG and QA can be useful for assessing the semantic adequacy of generated texts in RDF-to-Text tasks in English. Contemporaneous work by (Cohen et al., 2023) also similarly leverage consistency measures to verify the factuality of a model's generated answers. Nonetheless, these works remain in one modality and language only (English texts).

## 9 Conclusion

In this work, we examine the task of multimodal QG and QA from text and from graph in multiple languages which has application in QA-based evaluation of text generated from KG. By generating synthetic QA/QG data in English that has questions and answers aligned between text and graph, and using MT, heuristics and quality filters, we obtain "silver" data for Brazilian Portuguese and Russian that enables the training of multimodal multi-task models. Our models provide wider QG coverage, is cross-lingual for QG-QA from KG graphs, and achieves greater internal and cross-modal QA consistency over a baseline that is derived from work in English (Data-QuestEval) (Rebuffel et al., 2021) recently shown to be useful as a metric for evaluating the semantic consistency of RDF-to-Text generations. In fact, using our approach leads to > 12 points gain in the metric's correlation with human judgments for Russian. We also see strong performance in QA accuracy of up to 15.5 BSc when our approach's questions are posed to a multilingual retrieval-based QA model. Holistically, our approach's consistently better performance in coverage and QA consistency, demonstrates the improvements that it brings over the baseline.

## 10 Acknowledgments

We thank the anonymous reviewers for their feedback. This research was supported by ANR Project QUANTUM (Project-ANR-19-CE23-0025). We gratefully acknowledge the support of the French National Research Agency (Gardent; award ANR-20-CHIA-0003, XNLG "Multi-lingual, Multi-Source Text Generation") and of Facebook AI Research (FAIR) Paris. Experiments presented in this paper were carried out using the Grid'5000 testbed, supported by a scientific interest group hosted by Inria and including CNRS, RENATER and several Universities as well as other organizations (see https://www.grid5000.fr).

## 11 Limitations

Our data generation process relies on the KELM dataset (Agarwal et al., 2021), which was generated using a single pretrained language model (T5-large). As such, KELM — and hence the texts in our generated data (Q-KELM$^{En}$) too — reflects only the set of factual or linguistics characteristics in this model. For instance, there is certainly more than one way a KG graph (or set of facts) can be lexicalised in text, however KELM only contains one lexicalisation for each of the KG graph within it. In addition, we only use a single QG model for generating our initial sets of synthetic QA pairs Q-KELM$^{En}$. Although by using Q-KELM$^{En}$ to train our two general QG models, we may have introduced new varieties of questions into Q-WEBNLG$^{En}$ (and therefore Q-WEBNLG$^{PtBr}$ and Q-WEBNLG$^{Ru}$), it is unlikely we have obtained the full range of questions possible for a given context-answer pair. This limitation of KELM may however be alleviated by for example, ensembling the KELM dataset with generations from different LMs on the same subgraphs used to generate the KELM sentences/passages.

When setting up the KG QA training instances from Q-WEBNLG$^{PtBr}$/Q-WEBNLG$^{Ru}$ for complex questions, we chose quality over quantity and only used those originating from KG graphs (i.e. we excluded complex questions originated from texts).[11] This is because answers for questions generated from text may not be constrained to a single KG entity. This results in there being more textual QA instances than KG QA. For balance, it may be possible to using beam search (or variants of it such as diverse beam search (Vijayakumar et al., 2016) and constrained beam search (Post and Vilar, 2018)) to increase the generation of complex questions from graph using the Q-KELM$^{En}$-trained general purpose QG model. Finally, although we used the agreement of two state-of-the-art QA models when checking for QG acceptability, we cannot be certain that questions rejected by the QA models are not actually valid questions — in such cases, it means that the coverage of QA pairs in our datasets is constrained.

## 12 Ethics Statement

As advancements in generative technologies accelerate in terms of capabilities, scale and public access, so too must the need for the the ability to understand if such machine-generated information are reliable.

We believe that our multilingual KG/NL-QA/QG data creation method and multimodal multi-task QG-QA modeling has the potential to contribute positively in the following manner: (i) a cross-lingual (English-Brazilian Portuguese/Russian) QG-QA model with cross-modal consistency can aid in tasks that includes but are not limited to automated fact verification, KG-to-text/text-to-KG quality estimation and KG completion; and (ii) the ability to generate multilingual in-domain QA data can help improve downstream QA performance and dialogue systems to aid human-machine interactions with KGs in English as well as other languages. On the other hand, a direct application of our method and model for a task such as fact verification could lead to a failure to capture misinformation, which have the potential for substantive societal harm. This risk arises because KELM is based on a snapshot of the Wikidata KG from circa 2019. Additionally, the T5 pre-trained language models used in producing KELM, and also the mT5 pretrained checkpoints used in all our models, were trained with data up to 2020/2021. These constrain the extent and validity of factual knowledge in our model up to these points in time.

---

[11]Note that, without this step, our KG QA model would be steered towards generating Text QA-like answers. This would then bias the cross-modal comparisons (i.e. X-mod (GT) and X-mod (Gen Ans) and will also likely negatively affect the reliability of the Data-QuestEval metric to accurately assess the presence of KG facts mentioned in a given text.

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

## A Data details

### A.1 Synthetic QA data in English

We used a subset of KELM filtered for $(g, t)$ pairs where $g$ has between 2 and 5 triples; this is because larger sizes typically lead to unnatural questions. KELM (Agarwal et al., 2021) has 15M $(g, t)$ pairs of which we used a subset of about 2M pairs to seed the generation of synthetic QA data on; about 1.1M texts remain in Q-KELM$^{En}$ after our QA pair filtering steps. The distribution of Q-KELM$^{En}$'s questions by $nf$ can be found in Table 1.

### A.2 Synthetic QA data in Portuguese and Russian

**Quality filters for machine translation**

■ **Verifying MT on Portuguese texts** To ensure the quality of the machine translations of the texts from English to Portuguese, we backtranslated them and used two automatic scores: (i) BERTScore (BSc) (Zhang et al., 2020) and (ii) Google BLEU (GLEU) (Wu et al., 2016) to assess the semantic and lexical similarity of the translations and the original text — we used cut-offs of $\geqslant 0.7$ for BERTScore and $\geqslant 0.3$ for GLEU. We also used a regular expression to check for the presence of 10 or more consecutively repeated words which is a typical error in MT, indicating issues with the quality of the translation.

■ **Verifying MT on questions** For questions, we also use the same BSc and GLEU cutoffs above ($\geqslant 0.7$ and $\geqslant 0.3$) on the backtranslations of the questions to filter out poor quality translations.

### A.2.1 Aligning text answer to graph entity

The process here involves mapping $g$ to Portuguese/Russian using Wikidata multilingual labels for entity names. If a multilingual label for the language cannot be found, the name is translated from English using Google Translate API; here, we found that a commercial-grade MT system can be better suited to translate entity names. These multilingual versions of $g$ are only used to aid the mapping of $a_t^{PtBr}/a_t^{Ru}$ to a graph entity and are set aside thereafter (i.e. we do not use them in training)

## B Training and evaluation details

### B.1 Negative sampling for QA

For the textual QA and KG QA tasks, negative examples were created so as to allow the model to

| WebNLG Data | | | |
|---|---|---|---|
| Graph | [A] < Akita Museum of Art, floor count, 3 >
[B] < Akita Museum of Art, opening date, 2013-09-28 >
[C] < Akita Museum of Art, address, 1-4-2 Nakadori >
[D] < Akita Museum of Art, floor area, 3746.66 (sqm) > | | |
| Text | [E$^{En}$] The Akita Museum of Art at 142 Nakadori has 3 floors with a total area of 3746.66 square metres and was inaugurated on 28th September 2013. | [E$^{Ru}$] Акита Музей искусств по адресу 142 Накадори имеет 3 этажа общей площадью 3746,66 квадратных метров и был открыт 28 сентября 2013 года. | [E$^{PtBr}$] O Museu de Arte de Akita em 142 Nakadori tem 3 andares com uma área total de 3746,66 metros quadrados e foi inaugurado em 28 de setembro de 2013. |
| **Q-W*** | | | |
| X = graph | [B], [C] | | |
| a$_x$ = g ent | Akita Museum of Art | | |
| | {What museum opened in 2013-09-28 in Nakadori?, What is the name of the museum that opened in 2013-09-28 in Nakadori?} | {Какой музей открылся 28 сентября 2013 года в Накадори?, Как называется музей, открывшийся 28 сентября 2013 года в Накадори?} | {O que abriu o museu em 2013-09-28 em Nakadori?, Qual o nome do museu que se abriu em 2013-09-28 em Nakadori?} |
| X = graph | [B] | | |
| a$_x$ = g ent | 2013-9-28 | | |
| | {In what year was the Akita Museum of Art opened?, Which year was the Akita Museum of Art opened?} | {В каком году был открыт Художественный музей Акиты?, В каком году был открыт Художественный музей Акиты?} | {Em que ano foi aberto o Museu de Arte Akita?, O Museu de Arte de Akita foi inaugurado há algum ano?} |
| X = text | [E$^{En}$] | [E$^{Ru}$] | [E$^{PtBr}$] |
| a$_x$ = t span | {The Akita Museum of Art} | {Музей искусств Акита} | {Museu de Arte do Akita a 142 Nakadori} |
| | {What is the name of the museum that has 3 floors with a total area of 3746.66 square metres?} | {Как называется музей, который имеет 3 этажа и общую площадь 3746,66 квадратных метров?} | {Qual o nome do museu que possui 3 andares com área total de 3746,66 metros quadrados?} |
| X = text | [E$^{En}$] | [E$^{Ru}$] | [E$^{PtBr}$] |
| a$_x$ = t span | {2013} | {28 сентября 2013 года} | {2013} |
| | {What year was the Akita Museum of Art inaugurated? Which year was the museum inaugurated?} | {В каком году был открыт Художественный музей Акиты?, В каком году был открыт музей?} | {O Museu de Arte Akita foi inaugurado há algum ano?, Qual ano foi inaugurado o museu? } |

Table 7: Q-WEBNLG$^{En}$ instances derived from WebNLG Data and translated into Portuguese and Russian to give Q-WEBNLG$^{PtBr}$ and Q-WEBNLG$^{Ru}$. Enclosed letters refer to the triple/text above.

recognise questions that are unanswerable given the context. These samples are created using a random (i.e. 50-50) assignment to one of these two strategies:

■ **Strategy 1: simple negatives** for a given $(X, q, a_X)$ instance are created by picking another $(X_{other}, q', a_{X'})$ instance in the QA training set where $X$ and $X_{other}$ do not share any common entities/values between them. If $X$ is a text, we use the graph that it is paired with in WEBNLG to check for common entities. A new instance $(X, q', $ "unanswerable") is then created in the training data.

■ **Strategy 2: hard negatives** are created in the following manner: a mapping $M$ is first created ahead of time where every entity $e$ that can be found in the training data is associated with the set of all the $(X, q, a_X)$ instances where $e$ is mentioned in $X$ ($e \in X$). If $X$ is a text, we use the graph it is paired with in WEBNLG when creating $M$.

For a given $(X, q, a_X)$ instance, another instance, i.e. $(X_{other}, q', a_{X_{other}})$ is randomly chosen using $M$ and a random $e \in X$. If a single token from the answer $a_{X_{other}}$ overlaps with (i) any of the tokens for any $e$ (an entity or value) found in $X$ or (ii) any of the tokens of $X$, then this $(X_{other}, q', a_{X_{other}})$ candidate is rejected. Otherwise, a new instance $(X, q', $ "unanswerable") is created in the training data using the instance and the process for $(X, q, a_X)$ terminates. When a candidate instance

is rejected, a new $(X_{other'}, q', a_{X_{other'}})$ is drawn from $M$ using another $e \in X$. After 10 tries or when all $e \in X$ has been exhausted, a simple negative is created instead.

**Baseline: negative QA instances** For training the baseline's textual QA and KG QA models, we create unanswerable instances in a 2:1 ratio by randomly replacing the context for a question with another in the data.

## B.2 Upsampling

We carry out upsampling on two levels when preparing Q-WEBNLG$^{PtBr}$ and Q-WEBNLG$^{Ru}$ (the training data for our language specific multimodal multi-task QG-QA models).

■ **Between modalities for QG subtasks and between modalities for the same task** For QG this is done between complex textual and complex KG QG, simple textual and simple KG QG to ensure that the QG subtasks are balanced. It is also done between modalities for the QA tasks (e.g. textual QA and KG QA) to ensure that the tasks are balanced between modalities. The negative samples for the QA tasks are also upsampled in the same way.

For instance, $m_1$ and $m_2$ are the sets of samples for modality 1 and modality 2 respectively, and suppose $|m_1| > |m_2|$.

If $\mid m_2 \mid / \mid m_1 \mid \geqslant 0.5$, we randomly sample $\mid m_1 \mid - \mid m_2 \mid$ from $m_2$ to balance them.

If however $\mid m_2 \mid / \mid m_1 \mid < 0.5$, we upsample $m_2$ up to at most $1/3 \cdot \mid m_1 \mid$. This is to ensure that we do not overrepresent $m_2$ in the data and overfit on it during training.

■ **Globally for certain tasks**   This was done for simple QG as a whole (i.e. after simple textual and simple KG QG have been consolidated as one). This is because they are significantly less instantiated samples of this task in the data than the rest. The number of samples for these tasks were tripled.

### B.3   Evaluation settings

**BERTScore**   We use the same settings, except the following for a clearer analysis: (i) lowercasing, (ii) "unanswerable" strings were set to an empty string to avoid non-zero BSc for these and "over-counting" them, and (iii) rescaling BScs against a baseline (computed following the official method[12]) for a wider spread.

**Self-consistency filter**   Since the models are trained on machine-translated synthetic data, some generated questions may be ill-formed and pose an impact on QA. We therefore filter from both our model and the baseline, the questions: (i) which cannot be answered from their source context; or (ii) whose generated answer $\hat{a}_X$ has a BSc $< 0.7$ when compared against the reference $a_X$. We focus our analysis for QA Consistency (Section 4), the Data-QuestEval (Section 7.4) and mGEN (Section 7.5) evaluations on the results after filtering as this is the upper bound of the approaches' performance. For congruence with our QG Coverage analysis, if the filtering will leave a given approach A — and therefore B as well — with no QA pairs (i.e. no coverage), we keep one QA pair for A.

**Settings for mGEN experiments**   Since mGEN was trained with language tokens to produce answers in different languages for a question in language $L$, we leverage these tokens to replicate the same language setting for the inputs and outputs as our experimental set-up: (i) when the answering modality is a text, the input (both question and context) are in Portuguese/Russianand the output answer is in the same language. (ii) When the answering modality is a graph, we use the linearisation scheme from (Oguz et al., 2022) to utilise mGEN for KG QA. The question in the input is in Portuguese/Russian and the context is in English; the output answer is in English.

---

[12]https://github.com/Tiiiger/bert_score/blob/master/journal/rescale_baseline.md

# C Detailed results

## C.1 QA consistency: Token F1 and Exact Match

**Portuguese**

| QG QA | Internal | | X-Appr | |
|---|---|---|---|---|
| | Baseline / Baseline | QTT / QTT | Baseline / QTT | QTT / Baseline |
| **Same-mod (GT)** | | | | |
| T → T | $89.9_{(\pm 0.02)}$ | $94.9_{(\pm 0.25)}$ | $77.6^{(-12.3)}_{(\pm 0.09)}$ | $67.9^{(-27.0)}_{(\pm 0.26)}$ |
| G → G | $80.5_{(\pm (0.04))}$ | $90.4_{(\pm 0.31)}$ | $56.6^{(-23.9)}_{(\pm 0.11)}$ | $52.2^{(-38.2)}_{(\pm 0.26)}$ |

**Russian**

| QG QA | Internal | | X-Appr | |
|---|---|---|---|---|
| | Baseline / Baseline | QTT / QTT | Baseline / QTT | QTT / Baseline |
| **Same-mod (GT)** | | | | |
| T → T | $86.7_{(\pm 0.10)}$ | $84.7_{(\pm 0.14)}$ | $53.1^{(-33.6)}_{(\pm 0.06)}$ | $42.5^{(-42.2)}_{(\pm 0.11)}$ |
| G → G | $72.6_{(\pm 0.05)}$ | $95.2_{(\pm 0.27)}$ | $61.2^{(-11.4)}_{(\pm 0.15)}$ | $48.9^{(-46.3)}_{(\pm 0.58)}$ |

Table 8: **Consistency Results**. Average of **Token F1** between answers. In the first brackets () are the standard deviations across 5 random runs; for the right column, in the second brackets() are the difference between X-Appr and Internal.

**Portuguese**

| QG QA | Internal | | X-Appr | |
|---|---|---|---|---|
| | Baseline / Baseline | QTT / QTT | Baseline / QTT | QTT / Baseline |
| **Same-mod (GT)** | | | | |
| T → T | $72.5_{(\pm 0.08)}$ | $87.5_{(\pm 0.61)}$ | $58.1^{(-14.4)}_{(\pm 0.16)}$ | $43.2^{(-44.3)}_{(\pm 0.21)}$ |
| G → G | $73.2_{(\pm 0.10)}$ | $87.2_{(\pm 0.36)}$ | $48.9^{(-24.3)}_{(\pm 0.21)}$ | $41.9^{(-45.3)}_{(\pm 0.26)}$ |

**Russian**

| QG QA | Internal | | X-Appr | |
|---|---|---|---|---|
| | Baseline / Baseline | QTT / QTT | Baseline / QTT | QTT / Baseline |
| **Same-mod (GT)** | | | | |
| T → T | $58.3_{(\pm 0.23)}$ | $66.0_{(\pm 0.33)}$ | $33.8^{(-24.5)}_{(\pm 0.12)}$ | $36.6^{(-29.4)}_{(\pm 0.12)}$ |
| G → G | $60.9_{(\pm 0.16)}$ | $92.9_{(\pm 0.52)}$ | $39.3^{(-21.6)}_{(\pm 0.14)}$ | $33.9^{(-59.0)}_{(\pm 0.74)}$ |

Table 9: **Consistency Results.** Average of **Exact Match** between answers. In the first brackets () are the standard deviations across 5 random runs; for the right column, in the second brackets() are the difference between X-Appr and Internal

## C.2 Finer-grained QA consistency tests

**Portuguese**

| | Num Facts | | | |
|---|---|---|---|---|
| | 1 | 2 | 3 | 4 |
| **Same-mod (GT)** | | | | |
| T → T | 93.6 | 96.0 | 95.5 | 95.4 |
| G → G | 95.7 | 98.1 | 97.8 | 98.5 |
| **Cross-mod (GT)** | | | | |
| T → G | 65.6 | 62.5 | 58.2 | 54.6 |
| G → T | 67.1 | 66.0 | 61.0 | 57.0 |
| **Cross-mod (Gen Ans)** | | | | |
| T → G | 68.6 | 63.7 | 59.3 | 55.7 |
| G → T | 69.2 | 67.0 | 61.1 | 57.4 |

Table 10: **Fine-grained analysis of QTT's QA performance (BSc) for Portuguese. Num Facts** denote the number of facts ($nf$) the set of QA-pairs relate to (i.e. 1 denotes an SQ of 1 fact, 2 denotes a CQ of 2 facts etc...). The $nf$ sets are mutually exclusive.

**Russian**

| | Num Facts | | | |
|---|---|---|---|---|
| | 1 | 2 | 3 | 4 |
| **Same-mod (GT)** | | | | |
| T → T | 90.2 | 90.6 | 90.6 | 86.0 |
| G → G | 93.8 | 97.6 | 98.0 | 96.8 |
| **Cross-mod (GT)** | | | | |
| T → G | 29.8 | 30.1 | 30.8 | 29.5 |
| G → T | 29.6 | 29.3 | 30.0 | 31.0 |
| **Cross-mod (Gen Ans)** | | | | |
| T → G | 30.8 | 30.3 | 30.9 | 31.6 |
| G → T | 32.8 | 29.5 | 30.7 | 30.9 |

Table 11: **Fine-grained analysis of QTT's QA performance (BSc) for Russian. Num Facts** denote the number of facts ($nf$) the set of QA-pairs relate to (i.e. 1 denotes an SQ of 1 fact, 2 denotes a CQ of 2 facts etc...). The $nf$ sets are mutually exclusive.

## C.3 Multilingual Retrieval-based QA

**Portuguese**

| QG
QA | External Retrieval-QA | |
|---|---|---|
| | Baseline
mGEN | Ours
mGEN |
| | *Self (GT)* | |
| T → T | $73.1_{(\pm 0.04)}$ | $75.8_{(\pm 0.60)}$ |
| G → G | $56.3_{(\pm 0.10)}$ | $70.0_{(\pm 0.39)}$ |

**Russian**

| QG
QA | External Retrieval-QA | |
|---|---|---|
| | Baseline
mGEN | Ours
mGEN |
| | *Self (GT)* | |
| T → T | $61.7_{(\pm 0.19)}$ | $60.6_{(\pm 0.17)}$ |
| G → G | $59.4_{(\pm 0.17)}$ | $67.2_{(\pm 0.82)}$ |

Table 12: **QA consistency (Token F1)** comparing the QG of Baseline and Ours, using mGEN for QA.

**Portuguese**

| QG
QA | External Retrieval-QA | |
|---|---|---|
| | Baseline
mGEN | Ours
mGEN |
| | *Self (GT)* | |
| T → T | $52.4_{(\pm 0.13)}$ | $63.5_{(\pm 0.79)}$ |
| G → G | $45.2_{(\pm 0.15)}$ | $59.3_{(\pm 0.74)}$ |

**Russian**

| QG
QA | External Retrieval-QA | |
|---|---|---|
| | Baseline
mGEN | Ours
mGEN |
| | *Self (GT)* | |
| T → T | $36.4_{(\pm 0.13)}$ | $42.2_{(\pm 0.14)}$ |
| G → G | $45.8_{(\pm 0.20)}$ | $58.8_{(\pm 1.08)}$ |

Table 13: **QA consistency (Exact Match)** comparing the QG of Baseline and Ours, using mGEN for QA.