# OpenReview forum: "Multilingual Generation and Answering of Questions from Texts and Knowledge Graphs"
_EMNLP/2023/Conference — EMNLP 2023 Findings_

### Official Review · Reviewer_HqCQ · 2023-08-02

**Typos Grammar Style And Presentation Improvements:** Well written, except for few minor er…
**Soundness:** 3

**Excitement:**

4: Strong: This paper deepens the understanding of some phenomenon or lowers the barriers to an existing research direction.

**Missing References:**

None

**Paper Topic And Main Contributions:**

This paper introduces a multilingual (Brazilian Portuguese and Russian) and multimodal (knowledge graph and natural language) approach in the fields of Question Generation (QG) and Question Answering (QA), which is an important advance in research in this field.

Using synthetic data generation and machine translation methods, the authors generated QG-QA data aligned with graphs and texts to train multi-modal, multi-task models, and realized the translation of knowledge graphs and natural language texts in Portuguese and Russian. QG and QA. This approach provides an effective solution for QA evaluation using knowledge graphs to generate multilingual texts.

Experiments demonstrate that the proposed method outperforms a baseline approach derived from previous research on English and adapted to handle Portuguese and Russian.

**Questions For The Authors:**

1. In the process of data generation, it is mentioned that the proposed model only contains one lexical representation of each knowledge graph context. This may result in multiple lexical representations occurring in the text being ignored. For future research, is it possible to consider integrating the KELM dataset with the generation results of the same subgraph generated by other language models to expand the range of lexical representations covered?

2. When constructing KG QA training samples, how do the authors choose quality or quantity? It is recommended to give comprehensive examples for showing the new datasets.


3. Are there more methods for comparison？as the baseline comparison results given in the article are very limited. So, adding comparison methods can improve the persuasiveness of the article.
  4. In the experimental setting, the author chose quality instead of quantity, which is equivalent to biasing the experiment to their favorable situation, so when the number of experiments increases, I am curious whether the proposed method is scalable and robust.

**Reasons To Accept:**

1. The paper addresses an important problem in the field by extending QG-QA research to multilingual and multimodal scenarios.

2. The proposed method shows improvements over the baseline, suggesting its effectiveness in generating questions and providing answers in different languages and modalities.

3. The use of synthetic data generation and machine translation techniques to align graph-text data is innovative and contributes to the development of QG-QA models.

**Reasons To Reject:**

1. The evaluation of the proposed method could be further strengthened. It is recommended to provide more extensive experimental results, including additional metrics and comparisons against alternative approaches.

2. The limitations and potential biases of using synthetic data generation and machine translation are less discussed in the current version. Further analysis of their impact and potential challenges is needed to ensure the reliability of the proposed method.

3. The practical implications and applications of the proposed method for real-world scenarios should be discussed. Providing insights into its potential in the industry or specific domains would enhance the quality of the paper.

**Reproducibility:**

4: Could mostly reproduce the results, but there may be some variation because of sample variance or minor variations in their interpretation of the protocol or method.

**Reviewer Confidence:**

3: Pretty sure, but there's a chance I missed something. Although I have a good feel for this area in general, I did not carefully check the paper's details, e.g., the math, experimental design, or novelty.

---

> ### Author Rebuttal · Authors · 2023-08-26
>
> Thank you for your feedback and comments, we appreciate the opportunity to engage with you to discuss our work. To address your comments and questions:
>
> __Additional experimental validation of our approach__
>
> Regarding our evaluation approach, we wish to highlight that the results have to be seen holistically (i.e. (i) question coverage, (ii) self, cross-modal and cross-approach QA consistency, as well as (iii) downstream usage for the Data-QuestEval metric). Holistically, our approach’s consistently better performance provides strong support for the improvements that it brings over the baseline (one that we independently replicated from a recent approach with modifications for our multilingual setting).
>
> Nonetheless, we appreciate your (and Reviewer URPa’s) feedback on this aspect and have run further evaluation to provide additional validation for our approach. We do so in two ways:
>
> - Firstly, we extend our Section 7.3 (page 6) analysis by posing the questions generated by baseline and our approach to mGEN, a textQA model that is part of the CORA (Asai et al, 2021) state-of-the-art cross-lingual retrieval QA system (see below for details of experimental setup). This set of experiments can be seen as a second external validation (Line 425, page 7) of the QG quality (i.e. besides the cross-model X-Appr results in our submission) of baseline and our approach.
>
> - Secondly, we also add F1 and EM scores (see Tables 2 and 3 below) for the __Internal: Self (GT)__ comparison the same set of questions and answers as Table 4 of our submission (page 7), i.e. these new results will be placed next to the BSC figures we already reported. Since the __X-mod__ comparisons are cross-lingual and the F1 and EM would not be informative, we left these out.
>
> The additional mGEN experiments confirms our findings in the submission (Line 425, page 7); we observe meaningful increases (at least 2.6 and up to 15.1 BSC) in answer accuracy of our questions compared to baseline:
>
> > Portuguese:
> > - __+6.6__   __BSC__ 	for QGT -> QAT  (72.8 vs 57.7)
> > - __+15.1__ __BSC__ 	for QGG -> QAG (78.0 vs 71.4)
>
> > Russian:
> > - __+2.6__   __BSC__ 	for QGT -> QAT  (68.5 vs 65.9)
> > - __+9.1__   __BSC__ 	for QGG -> QAG (76.0 vs 66.9)
>
> Furthermore, our approach similarly outperforms baseline on (i) EM for all comparisons; and (ii) almost all cases in Token F1 (except for T -> T in Russian). These consistently strong findings with a second (alternative) QA approach should suffice in further demonstrating our approach’s substantial strength over the baseline.
>
> __Limitations and practical implications/applications__
>
> We wish to highlight that we do discuss the most pressing limitations and potential biases of the data in the Limitations section (Section 10, e.g. Line 555) and Ethics Statement (Section 11, e.g. Line 625, where we include an example of potential risk). We also discuss the relevance and applicability of our work in our Introduction and Ethics Statement (please see Lines 021 to 038, and 612). With the additional page for the camera-ready, we can raise these more prominently in the main body of the paper.
>
> __Responses to Questions__
>
> __Q1__: This will likely increase the lexical variety available in our synthetic QA/QG training data, but will however require additional resources to obtain, for e.g. some fine-tuning as was done in the original KELM paper, or possibly with few-shot in-context prompting on recent instruction-tuned models.
>
> __Q2__: We wish to highlight that we did provide example instances from the Q-WebNLG dataset, please see Table 6 in the appendix (page 13). To give an illustration of the rationale for this step, for e.g. a piece of text may be “The movie Z stars X and Y.” and its KG graph would be [ < X, stars in , Z >, < Y, stars in, Z > ]. In our Step 3 (Line 202), the set of possible answers for text are the noun phrases and named entities found in a given text, therefore one of the answers in this case could be “X and Y”, and a complex question that is generated from this answer and text might be “Who are the stars of Z?”, whose answer does not map to a single KG graph. Although we seek to maximise coverage available for training (Line 275), if we had simply used the questions generated from text (and their answers) as training instances for our KG QA, our model – which is generative – would learn to wrongly produce such answers for KG QA that would also “unfairly” bring them closer to the Text QA answers.
>
> __Q3__: As discussed above, we take your feedback into consideration and have included additional experimental findings above.
>
> __Q4__: We respectfully disagree with your comment that our prioritisation of “quality over quantity” (Line 587) for our KG QA training data unfairly biases the experimental set-up in our favour. To clarify:
>
> As our illustration for the answer to your Q2 shows, skipping this step would have the opposite effect, i.e. by steering our KG QA model towards generating Text QA-like answers for those cases, it will bias the cross-modal comparisons (i.e. X-mod (GT) and X-mod (Gen Ans) rows in Table 4, Section 7.3). Importantly, this will also likely negatively affect the reliability of the Data-QuestEval metric to accurately assess the presence of KG facts mentioned in a given text.
>
> In fact, we view our taking of this step as a potential limitation of our current work because the number of training instances for KG QA is less compared to the training instances available for Text QA and we had to rely on upsampling to balance the data (Line 975; needed since we are training in a multi-task manner where all tasks are present in a single training batch). It is possible that with better balance in KG QA data in the first place, it could possibly lead to further gains on the KG QA performance.
>
>
> ___
> We hope that these clarifications address your questions and concerns under Reasons to Reject, and that you will consider raising our score.
> ___
> ___
> \ \
>
>
> __Tables and references__
>
> | | |  |  |  |  |  |
> |---|:---:|:---:|:---:|:---:|:---:|:---:|
> | Portuguese | External QA |  |  |  |  |  |
> | QG | Baseline |  |  | Ours |  |  |
> | QA | mGEN |  |  | mGEN |  |  |
> | Modality | BSc | F1 | EM | BSc | F1 | EM |
> |  | Self (GT) |  |  |  |  |  |
> | T -> T | 71.4 (0.07) | 73.1 (0.04) | 52.4 (0.13) | 78.0 (0.58) | 75.8 (0.60) | 63.5 (0.79) |
> | G -> G | 57.7 (0.13) | 56.3 (0.10) | 45.2 (0.15) | 72.8 (0.48) | 70.0 (0.39) | 59.3 (0.74) |
> |---|:---:|:---:|:---:|:---:|:---:|:---:|
> | Russian | External QA |  |  |  |  |  |
> | QG | Baseline |  |  | Ours |  |  |
> | QA | mGEN |  |  | mGEN |  |  |
> | Modality | BSc | F1 | EM | BSc | F1 | EM |
> |  | Self (GT) |  |  |  |  |  |
> | T -> T | 65.9 (0.13) | 61.7 (0.19) | 36.4 (0.13) | 68.5 (0.18) | 60.6 (0.17) | 42.2 (0.14) |
> | G -> G | 66.9 (0.15) | 59.4 (0.17) | 45.8 (0.20) | 76.0 (0.77) | 67.2 (0.82) | 58.8 (1.08) |
>
> _Table 1: Downstream QA consistency results using mGEN as QA on questions generated by baseline and our approach. Average of automatic scores (BERTScore, Token F1 and Exact Match); in subscripts are the standard deviations across 5 random runs._
>
> | | | | | |
> |---|:---:|:---:|:---:|:---:|
> | Portuguese | Internal |  | X-Appr |  |
> | QG | Baseline | QTT | Baseline | QTT |
> | QA | Baseline | QTT | QTT | Baseline |
> | T -> T | 89.9 (0.02) | 94.9 (0.25) | 77.6 (0.09)(-12.3) | 67.9 (0.26)(-27.0) |
> | G -> G | 80.5 (0.04) | 90.4 (0.31) | 56.6 (0.11)(-23.9) | 52.2 (0.26)(-38.2) |
> | Russian | Internal |  | X-Appr |  |
> | QG | Baseline | QTT | Baseline | QTT |
> | QA | Baseline | QTT | QTT | Baseline |
> | T -> T | 86.7 (0.10) | 84.7 (0.14) | 53.1 (0.06)(-33.6) | 42.5 (0.11)(-42.2) |
> | G -> G | 72.6 (0.05) | 95.2 (0.27) | 61.2 (0.15)(-11.4) | 48.9 (0.58)(-46.3) |
>
> _Table 2: Consistency Results. Average of __Token F1__ between answers. In the first brackets () are the standard deviations across 5 random runs; for the right column, in the second brackets() are the difference between X-Appr and Internal._
>
> | | | | | |
> |---|:---:|:---:|:---:|:---:|
> | Portuguese | Internal |  | X-Appr |  |
> | QG | Baseline | QTT | Baseline | QTT |
> | QA | Baseline | QTT | QTT | Baseline |
> | T -> T | 72.5 (0.08) | 87.5 (0.61) | 58.1 (0.16)(-14.4) | 43.2 (0.21)(-44.3) |
> | G -> G | 73.2 (0.10) | 87.2 (0.36) | 48.9 (0.21)(-24.3) | 41.9 (0.26)(-45.3) |
> | Russian | Internal |  | X-Appr |  |
> | QG | Baseline | QTT | Baseline | QTT |
> | QA | Baseline | QTT | QTT | Baseline |
> | T -> T | 58.3 (0.23) | 66.0 (0.33) | 33.8 (0.12)(-24.5) | 36.6 (0.12)(-29.4) |
> | G -> G | 60.9 (0.16) | 92.9 (0.52) | 39.3 (0.14)(-21.6) | 33.9 (0.74)(-59.0) |
>
> _Table 3: Consistency Results. Average of __Exact Match__ between answers. In the first brackets () are the standard deviations across 5 random runs; for the right column, in the second brackets() are the difference between X-Appr and Internal._
>
> __Details for mGEN experiments__
>
> - mGEN was fine-tuned from a mT5-base checkpoint to generate the answer to a question given a collection of retrieved input contexts. We use the version (code and weights) that was released as part of the MIA-2022 Shared Task (Asai et al, 2022) https://github.com/mia-workshop/MIA-Shared-Task-2022/tree/main
> - The sets of questions posed to mGEN are the same as those for the results in our submission’s Table 4 (page 7), i.e. direct comparisons can be made between the additional results here and __X-Appr: Self (GT)__ results there.
> - Since mGEN was trained with language tokens to produce answers in different languages for a question in language L, we leverage these tokens to replicate the same language setting for the inputs and outputs as our experimental set-up:
>
>    - When the answering modality is a text, the input (both question and context) are in PtBr/Ru and the output answer is in the same language.
>
>    - When the answering modality is a graph (see Line 391 of our submission), we use the linearisation scheme from (Oguz et al., 2022) to utilise mGEN for KBQA. The question in the input is in PtBr/Ru and the context is in English; the output answer is in English.
>
> __References__
> 1. Asai et al, 2021: One Question Answering Model for Many Languages with Cross-lingual Dense Passage Retrieval
>
> 2. Asai et al, 2022: MIA 2022 Shared Task: Evaluating Cross-lingual Open-Retrieval Question Answering for 16 Diverse Languages
>
> 3. Oguz et al, 2022: UniK-QA: Unified Representations of Structured and Unstructured Knowledge for Open-Domain Question Answering

---

### Official Review · Reviewer_URPa · 2023-08-05

**Soundness:** 3

**Excitement:**

3: Ambivalent: It has merits (e.g., it reports state-of-the-art results, the idea is nice), but there are key weaknesses (e.g., it describes incremental work), and it can significantly benefit from another round of revision. However, I won't object to accepting it if my co-reviewers champion it.

**Paper Topic And Main Contributions:**

This paper proposes a framework for training multilingual, multimodal (natural language and graph), multi-task (QA and QG) models. The authors leverage synthetic data generation and machine translation to produce graph-text aligned QG-QA data in Portuguese and Russian. Experimental results show that the proposed method improves question coverage, QA consistency, and correlation to human judgements.

**Reasons To Accept:**

- This paper proposes a framework for training multilingual, multimodal (natural language and graph), multi-task (QA and QG) models. This is important as it provides a framework for languages with less resource.
- A well-thought framework that incorporates various modules of different purposes.

**Reasons To Reject:**

- Experimental results are a bit lacking. For example, are all the modules necessary?
- The presentation of the paper could be improved a lot. See comments below.

**Reproducibility:**

3: Could reproduce the results with some difficulty. The settings of parameters are underspecified or subjectively determined; the training/evaluation data are not widely available.

**Reviewer Confidence:**

2: Willing to defend my evaluation, but it is fairly likely that I missed some details, didn't understand some central points, or can't be sure about the novelty of the work.

**Typos Grammar Style And Presentation Improvements:**

### Comments

- Line 55, I would suggest ending this sentence with a colon without starting a new paragraph.
- The abstract, Section 1 and 2 are pretty hard to follow. What is the main research problem? Why is it important? I think the authors jump into the details too early and miss the opportunity to present the big picture of the paper.
- Since the framework consists of multiple tasks and modality, providing some examples would help readers understand better.

### Proofreading

- Line 552, are

---

> ### Author Rebuttal · Authors · 2023-08-26
>
> Thank you for your feedback on our work. We wish the provide the following response to your reasons to reject:
>
> __Experimental results and necessity of various modules__
>
> We take your (and Reviewer HqCQ’s) feedback into consideration and include additional experimental results. These include (i) an additional external validation of both baseline and our approach’s QG performance by using the mGEN reader from the CORA (Asai et al, 2021) state-of-the-art cross-lingual retrieval QA system, as well as (ii) additional token F1 and Exact Match scores. These results are consistent with the reported results in our submission and further show the gains from our approach’s over the baseline. For these results, please refer to the tables we provide in our response to Reviewer HqCQ.
>
> Since there is no cross-modal QG/QA data for Brazilian Portuguese and Russian, all of the modules are necessary for producing and ensuring the quality of the synthetic QG/QA data in these languages. The main modules (represented by the rectangle blocks in Figure 1, page 4) and their purpose are as follows:
>
> - _Off-the-shelf QG and QA models_: these are necessary to obtain an initial set of synthetic questions and answers (Line 173, page 3), as well as provide a check on answerability and consistency before using the initial data to train the intermediate controllable QG models (Line 209, page 3 and also see next point).
>
> - _Two controllable QG models that we trained (Line 194, page 3)_: these are also necessary because it is through controllable generation using them that we can significantly increase the QA/QG training data coverage (Line 207, page 3).
>
> - _Brazilian Portuguese and Russian machine translation and off-the-shelf QA (Line 226, page 3)_: these are necessary in order to obtain the questions as well as answers (and texts for Portuguese) in these languages. Note that since we are machine translating the questions from English, the semantics of the questions might be affected, which is why we posed the translated question to the PtBr/Ru TextQA, and verify the generated answer alignment to the original graph answer (Line 238, page 4).
>
> __Main research problem__
>
> Our main research problem is to be able to generate and answer questions across multiple modalities, from graph and from text, and to be able to do so for multiple languages besides English  (Line 044, page 1). To be useful, the approach has to be robust to surface variations, perform QA with accuracy and consistency, as well as generate wide QG coverage (Lines 056, 065 and 077, page 1). Note also that our work is on languages for which there is no available QA/QG training data that is aligned between the modalities. As we state in Introduction (Line 029, page 1), being able to do so can be particularly useful for verifying the consistency of information between modalities. As we also state in our Ethics Statement (Line 616, page 9), the areas where such verification can assist in include automatic fact verification (e.g. against a curated KG) as well as quality estimation of machine generated text from structured information (KG information), which are increasingly important as the prevalence of machine-generated is expected to increase. With the additional page available for the camera-ready, we can state these points more clearly in the abstract and Introduction.
>
> ___
> We hope that these clarifications address your concerns under Reasons to Reject, and that you will take them into consideration and raise our score.

---

### Official Review · Reviewer_zbSx · 2023-08-05

**Soundness:** 3

**Excitement:**

4: Strong: This paper deepens the understanding of some phenomenon or lowers the barriers to an existing research direction.

**Paper Topic And Main Contributions:**

The paper bridges QA and QG with multimodalities (text and graph) in a multilingual setting (Brazilian Portuguese and Russian).

**Questions For The Authors:**

A: Figures 2 and 3, do the authors check the quality of these generated questions?

B: Table 4 and the explanation doesn't seem to align.  Line 431: "... however, our approach’s stronger performance in all the Internal, X-mod, and X-Appr comparisons suggest that the former is the likely factor." but in Table 4 X-appr that doesn't seem to be the case. Is there perhaps a problem with the heading order?

**Reasons To Accept:**

The paper is the first one to attempt to do so in a multilingual setting.

It is shown to outperform the baseline technique (Rebuffel et al.'s) in some metrics.

**Reasons To Reject:**

The paper is an improvement to Rebuffel et al.'s work, but the approach is not novel and it comprises reused components from previous work.

Some results are confusing to follow. Some need further discussion to clarify the impact.

**Reproducibility:**

3: Could reproduce the results with some difficulty. The settings of parameters are underspecified or subjectively determined; the training/evaluation data are not widely available.

**Reviewer Confidence:**

2: Willing to defend my evaluation, but it is fairly likely that I missed some details, didn't understand some central points, or can't be sure about the novelty of the work.

**Typos Grammar Style And Presentation Improvements:**

`Figure 2` and `3`: be consistent either `Baseline` or `Rebuffel`. Fix the colors as well so the legend and the graph match; there seems to be some transparency issue.

---

> ### Author Rebuttal · Authors · 2023-08-26
>
> Thank you for your feedback and engaging with us to discuss our work.
>
> __Novelty of our approach__
>
> However, we respectfully disagree with your view that our work is not novel for the following reasons:
>
> - Our work provides an approach for generating synthetic data in two languages where such QA/QG training data that is aligned between text and graph modality does not exist. One of the languages is Brazilian Portuguese which is under-resourced.
>
> - This synthetic data enables the joint multi-modal multi-task training we propose, one that gives a single multi-modal multi-task QG+QA model (Section 6, page 4) that is significantly different from the approach in Rebuffel et al 2021 (that requires multiple models, i.e. one for each modality and task) which we derive the baseline from.
>
> - Importantly (for reasons we note in Line 077, page 1), our work – which is independent of theirs – departs significantly from the Rebuffel approach by also being able to maximise coverage (Line 275, page 4 and Line 319, page 5) via training to generate multiple questions from the same input in a single step.
>
> - Finally, to overcome the lack of any gold QA/QG data that is aligned between text and graph modality in these two languages, we designed our evaluation suite to include multiple internal, cross-modal and cross-approach checks (Line 341, page 6) to validate the performance of each approach. Note that we also show (Line 441, page 7) that these gains translate to improved correlations with human judgements over the baseline when used to compute the Data-QuestEval metric.
>
> __Question A: Quality of generated questions__
>
> None of the authors are speakers of these two languages, however, we wish to point out that our comparisons in Section 7.3 (particularly the cross-approach ones; page 6) give an indication of the quality of the questions generated by our approach over the baseline – see our discussion in Line 425 (page 7).
>
> Furthermore, we have also included additional experimental results in our response to Reviewer HqCQ; in summary (for details, please refer to the tables and the _“Additional experimental validation of our approach”_ section in our response there), we use another SOTA model for QA on the questions generated by baseline and our approach. The results there show that questions generated by our approach consistently lead to better QA accuracy compared to those generated by the baseline.
>
> __Question B: Understanding X-Appr results__
>
> The Table 4 (page 7) __X-Appr__ headings are correct. To interpret the X-Appr performance between baseline and our approach, we have to take the difference of the __X-Appr__ BSC score with the __Internal__ BSC score. This will give a fairer comparison since the QA performance is different between baseline and our approach.
>
> For instance, in the first row of Table 4 (page 7), the 73.6 BSC from QG (Baseline)+QA (Ours); T→T has to be compared against the QG (Baseline)+QA (Baseline); T→T setting on the left, giving the difference of -12.7 (in superscript brackets and in blue font).
>
> As seen from the table, whenever our QA is used (no matter the modality combination, or GT/Gen. Ans), the drop in QA accuracy is lower (in fact in half of the cases, there is even a gain).
>
> Based on your feedback, we agree that the table layout for the __X-Appr__ column can be improved and will modify it for the camera-ready to show the score difference in larger font instead of the raw BSC scores.
> ___
>
> We hope that these clarifications address your questions and concerns under Reasons to Reject, and you will consider raising your score.

---

### Meta-Review · Area_Chair_GN1U · 2023-09-17

**Recommendation:** 4

**Metareview:**

This paper presents an approach for multilingual question generation and question answering, comprising knowledge graphs and textual inputs. The paper is well positioned against the state of the art and can be seen as a (substantial, non-incremental) extension of Rebuffel et al.

The paper is solid and interesting. It addresses an important multilingual task and the model yields excellent results. Section 7 (Results) merits a special mention, as it discusses results from multiple different perspectives and offers a very good understanding of the system's behaviour. All the reviewers have appreciated the paper's "excitement".

However, the paper is extremely hard to read. It addresses three phenomena (multilinguality, kg/nlp and qg-qa) and thus offers a rather complex architecture with multiple rationales/subtasks/modules/.. I (AC) have some difficulty understanding individual parts -- and all the reviewers, I believe, have encountered the same: all the three reviewers have asked multiple clarification questions (addressed thoroughly by the authors in the rebuttal). I believe therefore that the presentation issue should be addressed, following very constructive comments: put more emphasis on the bigger picture (reviewer URPa), highlight use-case/impact (reviewer HqCQ), have a more principled discussion of modules and their roles (reviewers zbSx, URPa) etc.

---

### Decision · Program_Chairs · 2023-10-07

**Decision:**

Accept-Findings

**Comment:**

This paper presents an approach for multilingual question generation and question answering, comprising knowledge graphs and textual inputs. The paper is well positioned against the state of the art and can be seen as a (substantial, non-incremental) extension of Rebuffel et al.

The paper is solid and interesting. It addresses an important multilingual task and the model yields excellent results. Section 7 (Results) merits a special mention, as it discusses results from multiple different perspectives and offers a very good understanding of the system's behaviour. All the reviewers have appreciated the paper's "excitement".

However, the paper is extremely hard to read. It addresses three phenomena (multilinguality, kg/nlp and qg-qa) and thus offers a rather complex architecture with multiple rationales/subtasks/modules/.. I (AC) have some difficulty understanding individual parts -- and all the reviewers, I believe, have encountered the same: all the three reviewers have asked multiple clarification questions (addressed thoroughly by the authors in the rebuttal). I believe therefore that the presentation issue should be addressed, following very constructive comments: put more emphasis on the bigger picture (reviewer URPa), highlight use-case/impact (reviewer HqCQ), have a more principled discussion of modules and their roles (reviewers zbSx, URPa) etc.